# Immunosuppression variably impacts outcomes for patients hospitalized with COVID-19: A retrospective cohort study

Vijeeth Guggilla[1,2], Jennifer A. Pacheco[3], Alexandre M. Carvalho[2], Grant R. Whitmer[4], Anna E. Pawlowski[5], Jodi L. Johnson[6,7,8,9], Catherine A. Gao[10], Chad J. Achenbach[11,12], Theresa L. Walunas[1,4,6,9,11]*

1 Institute for Artificial Intelligence in Medicine, Northwestern University Feinberg School of Medicine, Chicago, Illinois, United States of America, 2 Northwestern University Feinberg School of Medicine, Chicago, Illinois, United States of America, 3 Center for Genetic Medicine, Northwestern University Feinberg School of Medicine, Chicago, Illinois, United States of America, 4 Department of Medicine, Northwestern University Feinberg School of Medicine, Chicago, Illinois, United States of America, 5 Northwestern Medicine Enterprise Data Warehouse, Northwestern University Feinberg School of Medicine, Chicago, Illinois, United States of America, 6 Robert H. Lurie Comprehensive Cancer Center of Northwestern University, Chicago, Illinois, United States of America, 7 Department of Pathology, Northwestern University Feinberg School of Medicine, Chicago, Illinois, United States of America, 8 Department of Dermatology, Northwestern University Feinberg School of Medicine, Chicago, Illinois, United States of America, 9 Department of Medical Social Sciences, Northwestern University Feinberg School of Medicine, Chicago, Illinois, United States of America, 10 Division of Pulmonary and Critical Care Medicine, Department of Medicine, Northwestern University Feinberg School of Medicine, Chicago, Illinois, United States of America, 11 Department of Preventive Medicine, Northwestern University Feinberg School of Medicine, Chicago, Illinois, United States of America, 12 Division of Infectious Diseases, Department of Medicine, Northwestern University Feinberg School of Medicine, Chicago, Illinois, United States of America

* t-walunas@northwestern.edu

## Abstract

### Background

Adults with immunosuppression are more likely to develop severe COVID-19 than adults without immunosuppression. Less is known about differences in outcomes for adults with immunosuppression who are hospitalized with COVID-19.

### Methods

A retrospective cohort study of adults hospitalized with COVID-19 at Northwestern Medicine hospitals between 03/01/2020 and 05/31/2022 was performed. Regression analyses were performed comparing in-hospital mortality, intensive care unit (ICU) admission, oxygenation requirements, and hospital/ICU length of stay among patients without immunosuppression (n = 9079) and patients with immunosuppression (n = 873).

**Data availability statement:** Data cannot be shared publicly because data contain protected health information (including identifying or sensitive patient information). Data cannot be shared publicly even if de-identified due to ethical restrictions from the Institutional Review Board related to risk of reidentification. Data are available from the corresponding author (t-walunas@northwestern.edu) or from the Northwestern Medicine Data Analytics team (researchanalytics@nm.org) upon reasonable request for researchers who meet the criteria for access to confidential data. Note that row-level data are access-controlled and cannot be provided publicly due to data use restrictions.

**Funding:** TLW received grant # CO-US-540-6535 from Gilead Sciences for this work. The Northwestern Medicine Enterprise Data Warehouse is supported by a grant from the National Institutes of Health, National Center for Advancing Translational Sciences, #UM1TR005121. The funders had no role in study design, data collection, analysis, decision to publish or preparation of the manuscript. Gilead Sciences did review the final manuscript.

**Competing interests:** The authors have declared that no competing interests exist.

## Results

Patients with immunosuppression had significantly higher mortality than patients without immunosuppression (OR: 1.33, 95% CI: 1.11–1.60). This effect was even stronger when controlling for age at admission, diabetes, obesity, SARS-CoV-2 variant era, and COVID-19 medication use (adjusted OR: 1.78, 95% CI: 1.46–2.16). ICU admission (adjusted OR: 1.64, 95% CI: 1.41–1.90) and invasive ventilation (adjusted OR: 1.68, 95% CI: 1.36–2.06) were also significantly higher in patients with immunosuppression. Hospitalization length (median: 7 days) and ICU length of stay (median: 2.5 days) were longer in patients with immunosuppression compared to patients without immunosuppression (median: 5 days, adjusted $p < 0.001$; median: 2 days, adjusted $p = 0.04$). Subgroup analyses showed that patients with solid organ transplant, HIV with low CD4 cell count, and secondary immunodeficiency had significantly higher adjusted mortality and ICU admission compared to patients without immunosuppression. Patients with solid organ transplant also had significantly higher invasive ventilation and ICU length of stay.

## Conclusions

Patients with immunosuppression had worse outcomes than patients without immunosuppression. Subgroup analyses showed that patients with solid organ transplant had the worst outcomes overall. Patients with HIV had similar outcomes as patients without immunosuppression unless CD4 cell count was low.

## Introduction

COVID-19 cases have substantially decreased thanks to vaccination and rising population immunity, but risks continue to be elevated for individuals with pre-existing immunosuppression. A systematic review comparing patients with solid organ transplant (SOT) to the general population hospitalized with COVID-19 showed higher rates of intensive care and in-hospital mortality in the SOT group [1]. HIV has also been identified as a risk factor for poor outcomes in the context of COVID-19. Multiple systematic reviews and cohort studies have found significantly higher COVID-19 associated mortality in people with HIV (PWH) compared to those without HIV [2–6]. Finally, individuals with primary (PI) and secondary immunodeficiencies (SI) in the United Kingdom Primary Immunodeficiency Network who developed COVID-19 as well as a cohort of individuals with PI in New York City with COVID-19 had higher inpatient mortality than the general population [7,8]. A systematic review performed by the Centers for Disease Control and Prevention (CDC) found similar results as well as higher rates of ICU admission for patients with PI who developed COVID-19 [9].While this evidence suggests that SOT, HIV, PI, and SI increase the risks associated with COVID-19, there is also evidence to the contrary. For example, three different cohort studies in the US, Spain, and Italy have found that compared to patients from the general population, patients with SOT hospitalized with COVID-19

had no differences in mortality, with the US-based study suggesting that differences in age and comorbidities rather than SOT status itself are contributors to any observed differences in mortality [10–12]. With regard to HIV, studies have shown increased mortality with COVID-19 co-infection, but there have also been a number of cohort studies in PWH hospitalized with COVID-19 that have shown no differences in mortality compared to those without HIV [13–26]. Overall, this suggests important differences in outcomes when comparing populations hospitalized with COVID-19.

Clearly, the immune milieu impacts COVID-19 progression and severity, but much remains to be understood about the impact of specific reasons for immunosuppression on outcomes of severe COVID-19 among hospitalized patients. Many of the cohort studies that have been performed have had relatively small sample sizes and have shown inconsistent results. Many were also performed on populations outside of the United States. While past systematic reviews have usually included many studies, they have also suffered from including small and imbalanced cohorts. Furthermore, past studies have left important gaps in our understanding of this domain. For example, there have been no studies comparing the outcomes of COVID-19 hospitalization in individuals with PI and SI, and the evidence surrounding COVID-19 hospitalization outcomes among patients with SOT has been inconsistent. While COVID-19 hospitalization outcomes have been extensively studied in PWH, the influence of low CD4 cell count on outcomes of COVID-19 has been less clear [14,18,24,25,27,28]. In addition, there have been no studies comparing all four of these reasons for immunosuppression in a single cohort study with a unified approach. Also, although many studies have focused on mortality, there are fewer studies comparing other COVID-19 hospital outcomes such as ICU care, ventilatory support, and hospitalization length. Finally, prior research on COVID-19 and immunosuppression occurred through 2020 or 2021, with few studies in 2022, the era of Omicron SARS-CoV-2 subvariants and high levels of vaccine-induced and natural immunity.

We aimed to address these gaps to improve our knowledge of COVID-19 epidemiology and outcomes for different populations with immunosuppression, which is crucial for public health guidance [29]. Many of these gaps exist because studying COVID-19 hospitalization outcomes for patients with specific reasons for immunosuppression has been challenging due to the rarity of individuals with such reasons for immunosuppression at a given healthcare institution. Given these challenges, leveraging data available in the electronic health record (EHR) is a robust strategy for studying how severe COVID-19 outcomes differ among individuals with different reasons for immunosuppression. In this study, we compared differences in in-hospital mortality, ICU admission, oxygenation requirements, and hospital/ICU length of stay between adults with a history of SOT, HIV, PI, or SI hospitalized with COVID-19 and adults without immunosuppression hospitalized with COVID-19.

## Methods

We report this article based on STROBE guidelines (S1 File) [30].

### Study design and data source

We performed a cohort study utilizing data from the Northwestern Medicine (NM) Enterprise Data Warehouse (NMEDW). The NMEDW is the clinical research database of NM, a large healthcare system providing inpatient, outpatient, and specialty care throughout Chicago and Northern Illinois. The NMEDW contains data, including diagnosis (ICD-9 and ICD-10 codes), hospitalization, treatment, laboratory, and demographic information for over 10 million patients who have received care at NM hospitals and clinics.

### Study population

We sampled patient data from the NMEDW. The inclusion criteria were receiving a positive COVID-19 laboratory test (positive polymerase chain reaction on nasopharyngeal or lower respiratory tract specimens, positive SARS-CoV-2 antigenic test) between 03/01/2020 and 05/31/2022, being hospitalized in the period encompassing one day before to seven days after the laboratory test result, and being at least 18 years of age at the time of hospitalization. The exclusion criteria

were not being hospitalized, testing positive for COVID-19 more than one day after hospitalization, and being hospitalized for less than two days. These exclusion criteria helped minimize inclusion of individuals hospitalized for less severe disease and individuals hospitalized for reasons other than COVID-19 who were quickly discharged. The study size was arrived at through these inclusion and exclusion criteria.

### Data collection

All data was extracted retrospectively from the NMEDW by power users of the NMEDW trained for data collection. Diagnosis codes that patients had received prior to their hospitalization with COVID-19 were extracted to assign each patient to either non-exposure (without immunosuppression) or one of four exposure subgroups. Absolute CD4 T cell counts closest to the time of hospitalization including the period encompassing seven days after hospitalization were also extracted for exposure subgroup assignment.

To assess study outcomes, dates of hospital and ICU admission and discharge, in-hospital oxygenation information, and in-hospital all-cause mortality were also extracted. To assess study covariates and sample demographics, age at admission, sex assigned at birth, race, ethnicity, most recent body mass index (BMI) measurement, diagnosis codes for diabetes, and in-hospital use of remdesivir and Paxlovid were extracted. Demographic variables were stratified to the same level of detail present in the NMEDW (i.e., sex assigned at birth is recorded as either Male or Female; ethnicity is recorded as either Hispanic or Latino, Not Hispanic or Latino, or Other). Bias during data collection was minimized by defining our study population broadly and extracting all relevant data available. Given the missingness inherent in electronic health record (EHR) data due to its real-world nature, we also chose data types known to have low missingness (e.g. admission dates, in-hospital mortality) to capture study outcomes.

### Exposure and outcomes

We compared COVID-19 hospitalization outcomes between the non-exposure (without immunosuppression) and exposure (with immunosuppression) groups. The exposure group was further divided into the following subgroups: SOT (includes kidney transplant, liver transplant, heart transplant, and lung transplant), HIV without low CD4 (absolute CD4 T cell count > 350 cells/μL), HIV with low CD4 (absolute CD4 T cell count <= 350 cells/μL), PI (includes antibody deficiency, combined immunodeficiency, common variable immunodeficiency, ataxia-telangiectasia, phagocyte deficiency, and complement deficiency), or SI (includes patients taking immunosuppressive medication for autoimmune disease and patients who have received hematopoietic stem cell transplants). We defined SI in this way to capture common reasons for immunosuppression which we noted in our cohort that did not fall into the other groups we identified. This also aligned with representation of SI in the literature. We selected these subgroups to specifically address the inconsistencies and gaps in the literature regarding outcomes for these groups.

All patients assigned to an immunosuppression subgroup were manually adjudicated to accurately classify categorization of their immunosuppression. Patients found to have a different immunosuppression subgroup than expected based on ICD coding were reassigned to the appropriate group. Patients found to fall into multiple subgroups of immunosuppression were assigned to only one group per the following hierarchy: SOT, HIV, PI, and SI. Manual adjudication was guided by clinical experts and performed either by physicians or medical students with physician support.

The primary outcome was in-hospital mortality. Secondary outcomes included hospitalization length, oxygenation requirements, and ICU admission/length of stay. These outcomes were chosen to not only understand the overall impact of immunosuppression and compare results to previous studies, but also to provide a more granular understanding of severity differences due to immunosuppression. There was no follow-up beyond hospitalization. Dates of admission and discharge were used to determine hospitalization length. Rate of ICU admission and length of ICU stay were determined based on dates of ICU admission and discharge. For patients who were also admitted to the ICU, hospitalization length includes time spent in the ICU. Oxygenation requirements were categorized as use of low flow oxygen, non-invasive

ventilation (including high-flow nasal cannula), and invasive ventilation (includes intubation and ECMO). Dates of hospital admission were also used to assign SARS-CoV-2 variant era with categorization into three groups based on calendar time of when a specific variant of concern (VOC) accounted for greater than 50% of circulating viruses determined per sequencing surveillance. The eras falling within our study timeline were defined as: Pre-Delta (ancestral and early variants of concern including alpha and gamma, 03/01/2020 to 06/30/2021), Delta (07/01/2021 to 12/31/2021), and Omicron (01/01/2022 to 05/31/2022).

## Statistical analysis

Multivariate logistic regression analyses were performed in R adjusting for age at admission, diabetes, obesity, assigned SARS-CoV-2 variant era, and COVID-19 medication use to assess the effect of exposure on in-hospital mortality, administration of low flow oxygen, non-invasive ventilation, and invasive ventilation, and ICU admission. The effect of exposure subgroup was assessed in the same way. We chose to adjust for these covariates because age, diabetes, and obesity have all been identified as the strongest risk factors for death in patients hospitalized with COVID-19 and because of changes in pandemic landscape over time with the rise and fall of different VOCs and the introduction of new treatment options [31,32]. We did not apply competing risks analysis when assessing these binary variables. For all logistic regression analyses, influential values were assessed by isolating any observations with a standardized residual greater than three and performing sensitivity analyses. The variance inflation factor was used to confirm that there was no multicollinearity observed across all the analyses. Binary outcomes are reported as frequencies. Comparison results are reported as unadjusted and adjusted odds ratios with 95% confidence intervals.

Multivariate linear regression analyses adjusting for the same covariates were performed in R to assess the effect of exposure on hospitalization length and ICU length of stay. The effect of exposure subgroup was assessed in the same way. For both hospitalization length and ICU length of stay, stratified analyses of time to either discharge alive or time to in-hospital mortality were also performed. For hospitalization length, the logarithm transform of the data was used to achieve normality of the residuals and back-transformation of estimates from the log residual model were produced using exponentiation via the emmeans R package. Median and interquartile range were used to report summary statistics. Comparison results are reported as unadjusted and adjusted odds ratios with 95% confidence intervals.

To account for competing risks when comparing hospitalization length and ICU length of stay, time-to-event analyses were also performed with discharge alive as the event of interest and in-hospital mortality as the competing event. Gray's test was used to compare the cumulative incidence functions of discharge between exposure subgroups, and a Fine-Gray proportional sub-distribution hazards model was implemented to examine the association between exposure subgroup and discharge incidence adjusting for the same covariates.

For all analyses, the threshold for significance was set at 5% with 95% confidence intervals. There were no missing observations. Full regression analysis results are in S1 Table. All analyses were performed in the R software environment, Version 4.3.1 (2023-06-16).

## Ethics statement

This was a retrospective study performed under a waiver of consent. This study was approved by the Northwestern University Institutional Review Board (STU00217318), which operates under the principles of the Declaration of Helsinki.

## Results

### Characteristics of the study population at baseline

We identified 10713 adult patients hospitalized with COVID-19 at NM hospitals between 03/01/2020 and 05/31/2022. Of these, 9952 patients were included in the study for being hospitalized for at least two days. Of these, 1123 patients were

manually adjudicated and confirmed to have been hospitalized with COVID-19. After further categorization based on ICD codes and CD4 T cell count (for PWH) and manual adjudication, we analyzed 9079 adult patients without immunosuppression, 473 adult patients with SOT, 41 adult PWH without low CD4 T cell count, 42 adult PWH with low CD4 T cell count, 17 adult patients with PI, and 300 adult patients with SI (S1 Fig). Complete data were available for the demographic variables and outcomes we assessed. Patients were not followed-up beyond hospitalization.

The median age of the total included cohort was 63 years. Compared to the non-exposure (without immunosuppression) group, the exposure (with immunosuppression) group was significantly younger (median: 59 years exposure, 64 years non-exposure, p-value < 0.001). Of the exposure subgroups, PWH without and with low CD4 T cell count had the lowest median ages. Among the included cohort, 47.3% were assigned male and 52.7% were assigned female at birth. The proportion of men was significantly higher in the exposure group (58.7%) compared to the non-exposure group (46.2%, p-value < 0.001). Similar proportions of men to women were seen in patients with PI, but 59.4% and 63.4% of patients with SOT and PWH without low CD4 T cell count were men, respectively. There were also significant differences in race, SARS-CoV-2 variant era, BMI, diabetes, and remdesivir use between patients in the non-exposure and exposure groups (p-value < 0.001 for all comparisons). Complete descriptive statistics of the COVID-19 study population and its respective groups are shown in Table 1.

### In-hospital mortality

The primary outcome we assessed was in-hospital mortality (Fig 1). Patients in the exposure (with immunosuppression) group had significantly higher mortality (OR: 1.33, 95% CI: 1.11–1.60) than patients in the non-exposure (without immunosuppression group). This effect was even stronger when adjusting for age at admission, diabetes, obesity, SARS-CoV-2 variant era, and COVID-19 medication use (adjusted OR: 1.78, 95% CI: 1.46–2.16). We adjusted for these covariates because age, diabetes, and obesity have all been identified as the strongest risk factors for death in patients hospitalized with COVID-19, and because different variants and treatment options are known to influence COVID-19 outcomes. Subgroup analysis showed that patients with SOT (adjusted OR: 1.76, 95% CI: 1.34–2.27), PWH with low CD4 (adjusted OR: 3.21, 95% CI: 1.41–6.63), and patients with SI (adjusted OR: 1.86, 95% CI: 1.36–2.49) had significantly higher mortality than patients without immunosuppression. There were no significant differences in mortality for PWH without low CD4 and patients with PI compared to patients without immunosuppression (Table 2). Similar trends were observed when stratifying results by sex, except that in males, there were no significant differences in mortality for PWH with low CD4 and patients with SI compared to patients without immunosuppression (S2 and S3 Tables).

### ICU admission

Our study only included hospitalized patients, and many of these patients were also admitted to the ICU while hospitalized. Similar trends to in-hospital mortality were observed for rates of ICU admission. Patients with immunosuppression had significantly higher rates of ICU admission (adjusted OR: 1.64, 95% CI: 1.41–1.90) compared to patients without immunosuppression when adjusting for age at admission, diabetes, obesity, SARS-CoV-2 variant era, and COVID-19 medication use. Again, subgroup analysis showed that patients with SOT (adjusted OR: 1.71, 95% CI: 1.40–2.07), PWH with low CD4 (adjusted OR: 1.97, 95% CI: 1.03–3.67), and patients with SI (adjusted OR: 1.68, 95% CI: 1.31–2.13) had significantly higher rates of ICU admission compared to patients without immunosuppression. PWH without low CD4 and patients with PI did not have significantly different rates of ICU admission compared to patients without immunosuppression (Table 3). Sex stratified results are available in S2 and S3 Tables.

### Oxygenation requirements

We also assessed oxygenation requirements among patients without immunosuppression and the exposure subgroups, focusing on the use of low flow oxygenation, non-invasive ventilation, and invasive ventilation. While there

**Table 1. Baseline characteristics of the study population.**

| | Total | Non-exposure (without immu-nosuppression) | Exposure (with immuno-suppression) | p-value | Solid organ transplant | HIV without low CD4 | HIV with low CD4 | Primary immuno-deficiency | Secondary immuno-deficiency |
|---|---|---|---|---|---|---|---|---|---|
| | n = 9952 | n = 9079 | n = 873 | | n = 473 | n = 41 | n = 42 | n = 17 | n = 300 |
| **Median age at admission, y (IQR)** | 63.00 (48.00, 76.00) | 64.00 (48.00, 77.00) | 59.00 (48.00, 67.00) | < 0.001 | 58.00 (47.00, 66.00) | 52.00 (40.00, 62.00) | 51.00 (45.00, 61.75) | 61.00 (52.00, 68.00) | 62.00 (53.00, 71.00) |
| **Sex assigned at birth, n (%)** | | | | < 0.001 | | | | | |
| Men | 4705 (47.28) | 4193 (46.18) | 512 (58.65) | | 281 (59.41) | 26 (63.41) | -- | -- | 162 (54.00) |
| Women | 5247 (52.72) | 4886 (53.82) | 361 (41.35) | | 192 (40.59) | 15 (36.59) | -- | -- | 138 (46.00) |
| **Race, n (%)** | | | | < 0.001 | | | | | |
| American Indian or Alaska Native | 39 (0.39) | 34 (0.37) | -- | | -- | 0 (0.00) | 0 (0.00) | 0 (0.00) | -- |
| Asian | 336 (3.38) | 302 (3.33) | 34 (3.89) | | 23 (4.86) | -- | -- | 0 (0.00) | 9 (3.00) |
| Black or African American | 1682 (16.90) | 1462 (16.10) | 220 (25.20) | | 112 (23.68) | 26 (63.41) | 19 (45.24) | -- | 62 (20.67) |
| Native Hawaiian or Other Pacific Islander | 31 (0.31) | 29 (0.32) | -- | | -- | 0 (0.00) | 0 (0.00) | 0 (0.00) | -- |
| White | 6710 (67.42) | 6214 (68.44) | 116 (13.29) | | 257 (54.33) | 13 (31.71) | 17 (40.48) | 13 (76.47) | 196 (65.33) |
| Unknown | 1154 (11.60) | 1038 (11.43) | 496 (56.82) | | 76 (16.07) | -- | -- | -- | 31 (10.33) |
| **Ethnicity, n (%)** | | | | 0.24 | | | | | |
| Hispanic or Latino | 1960 (19.69) | 1770 (19.50) | 190 (21.76) | | -- | -- | -- | -- | -- |
| Not Hispanic or Latino | 7758 (77.95) | 7097 (78.17) | 661 (75.72) | | 328 (69.34) | 36 (87.80) | 34 (80.95) | 14 (82.35) | 249 (83.00) |
| Unknown | 234 (2.35) | 212 (2.34) | 22 (2.52) | | -- | -- | -- | -- | -- |
| **SARS-CoV-2 variant era, n (%)** | | | | < 0.001 | | | | | |
| Pre-Delta | 5826 (58.54) | 5363 (59.07) | 154 (17.64) | | 241 (50.95) | 26 (63.41) | 26 (61.90) | 10 (58.82) | 160 (53.33) |
| Delta | 2039 (20.49) | 1885 (20.76) | 250 (28.64) | | 86 (18.18) | -- | -- | -- | 49 (16.33) |
| Omicron | 2035 (20.45) | 1785 (19.66) | 463 (53.04) | | 144 (30.44) | -- | -- | -- | 87 (29.00) |
| **Median BMI (IQR)** | 29.29 (25.11, 34.74) | 29.45 (25.23, 34.97) | 27.94 (24.28, 32.44) | < 0.001 | 27.88 (24.44, 31.70) | 27.87 (23.44, 34.21) | 24.56 (21.33, 29.07) | 27.66 (26.17, 30.08) | 28.61 (24.51, 33.72) |
| **Diabetes, n (%)** | 3553 (35.70) | 3161 (34.82) | 392 (44.90) | < 0.001 | 264 (55.81) | 18 (43.90) | 10 (23.81) | -- | 96 (32.00) |
| **Remdesivir use, n (%)** | 4164 (41.84) | 3675 (40.48) | 489 (56.01) | < 0.001 | 275 (58.14) | 14 (34.15) | 20 (47.62) | 13 (76.47) | 167 (55.67) |
| **Paxlovid use, n (%)** | 2 (0.02) | 2 (0.02) | 0 (0.00) | > 0.99 | 0 (0.00) | 0 (0.00) | 0 (0.00) | 0 (0.00) | 0 (0.00) |

Counts less than 10 and greater than zero are not reported (denoted by "--") to protect patient privacy.

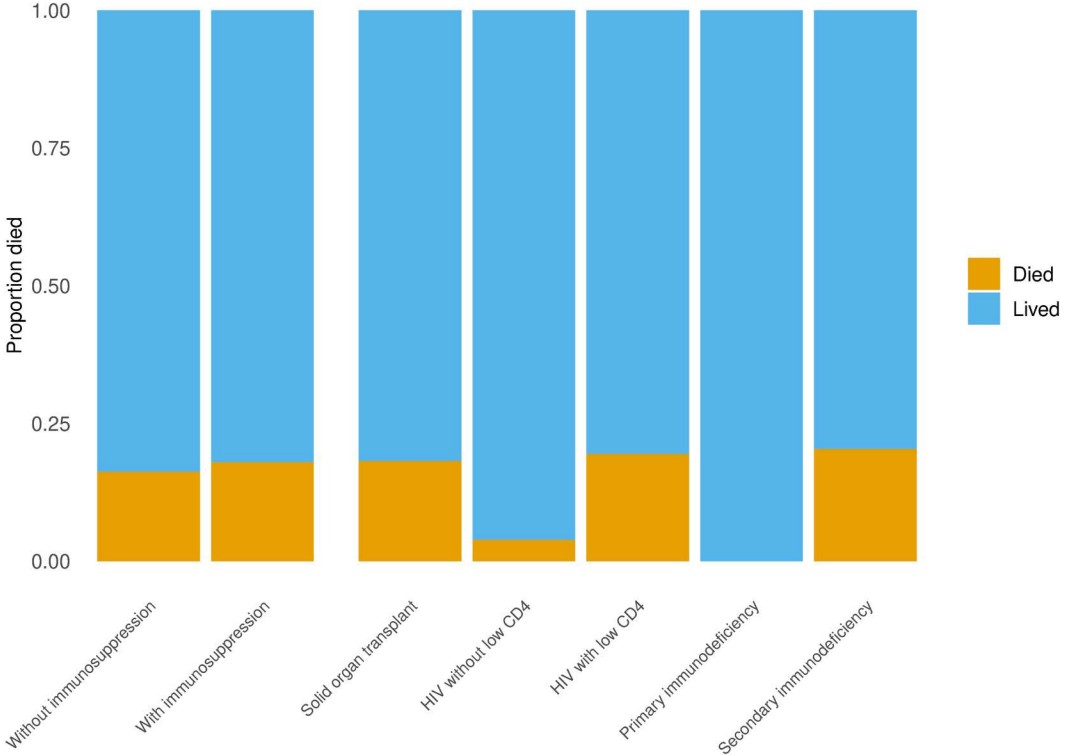

**Fig 1. In-hospital mortality of the COVID-19 study population across groups.** In-hospital mortality was extracted from the electronic health record data available for each patient. In-hospital mortality for the entire non-exposure (without immunosuppression) and exposure (with immunosuppression) groups is shown, followed by in-hospital mortality for the exposure subgroups.

**Table 2. In-hospital mortality of the COVID-19 study population across groups.**

| | Non-exposure | Exposure | Solid organ transplant | HIV without low CD4 | HIV with low CD4 | Primary immunodeficiency | Secondary immu-nodeficiency |
|---|---|---|---|---|---|---|---|
| | n = 9079 | n = 873 | n = 473 | n = 41 | n = 42 | n = 17 | n = 300 |
| **In-hospital mortality, n (%)** | 1400 (14.07) | 153 (17.53) | 79 (16.70) | 3 (7.32) | 9 (21.43) | 1 (5.88) | 61 (20.33) |
| **Unadjusted OR (95% CI)** | Ref | **1.33 (1.11, 1.60)** | 1.26 (0.98, 1.61) | 0.50 (0.12, 1.37) | 1.71 (0.77, 3.44) | 0.39 (0.02, 1.92) | **1.60 (1.19, 2.12)** |
| **Adjusted OR (95% CI)** | Ref | **1.78 (1.46, 2.16)** | **1.76 (1.34, 2.27)** | 0.90 (0.21, 2.55) | **3.21 (1.41, 6.63)** | 0.45 (0.02, 2.27) | **1.86 (1.36, 2.49)** |

Adjusted odds ratios are from multivariate regression analyses with age at admission, diabetes, obesity, SARS-CoV-2 variant era, remdesivir use, and Paxlovid use as covariates with non-exposure (without immunosuppression) as the reference. Significant odds ratios are in bold.

was no significant difference in low flow oxygenation between patients without immunosuppression and patients with immunosuppression, subgroup analysis showed significantly greater use in patients with PI (OR: 5.76, 95% CI: 1.17–103.92) and patients with SI (OR: 1.38, 95% CI: 1.05–1.85). However, these effects were not maintained when adjusting for age at admission, diabetes, obesity, SARS-CoV-2 variant era, and COVID-19 medication use (Table 4).

For non-invasive ventilation, there was no significant difference between patients without immunosuppression and patients with immunosuppression when adjusting for the same covariates. However, in the adjusted subgroup analysis,

**Table 3. ICU admission rates of the COVID-19 study population across groups.**

| | Non-exposure | Exposure | Solid organ transplant | HIV without low CD4 | HIV with low CD4 | Primary immunodeficiency | Secondary immunodeficiency |
|---|---|---|---|---|---|---|---|
| | n = 9079 | n = 873 | n = 473 | n = 41 | n = 42 | n = 17 | n = 300 |
| Admitted to ICU, n (%) | 2757 (27.70) | 335 (38.37) | 191 (40.38) | 9 (21.95) | 16 (38.10) | 4 (23.53) | 115 (38.33) |
| Unadjusted OR (95% CI) | Ref | **1.71 (1.48, 1.98)** | **1.86 (1.54, 2.25)** | 0.77 (0.35, 1.55) | 1.69 (0.89, 3.13) | 0.85 (0.24, 2.39) | **1.71 (1.34, 2.16)** |
| Adjusted OR (95% CI) | Ref | **1.64 (1.41, 1.90)** | **1.71 (1.40, 2.07)** | 0.83 (0.37, 1.68) | **1.97 (1.03, 3.67)** | 0.82 (0.23, 2.35) | **1.68 (1.31, 2.13)** |

Adjusted odds ratios are from multivariate regression analyses with age at admission, diabetes, obesity, SARS-CoV-2 variant era, remdesivir use, and Paxlovid use as covariates with non-exposure (without immunosuppression) as the reference. Significant odds ratios are in bold.

**Table 4. Oxygenation requirements of the COVID-19 study population across groups.**

| | Non-exposure | Exposure | Solid organ transplant | HIV without low CD4 | HIV with low CD4 | Primary immunodeficiency | Secondary immunodeficiency |
|---|---|---|---|---|---|---|---|
| | n = 9079 | n = 873 | n = 473 | n = 41 | n = 42 | n = 17 | n = 300 |
| Low flow oxygenation, n (%) | 7337 (73.72) | 660 (75.60) | 353 (74.63) | 27 (65.85) | 26 (61.90) | 16 (94.12) | 238 (79.33) |
| Unadjusted OR (95% CI) | Ref | 1.11 (0.95, 1.31) | 1.06 (0.86, 1.31) | 0.69 (0.37, 1.36) | 0.58 (0.32, 1.11) | **5.76 (1.17, 103.92)** | **1.38 (1.05, 1.85)** |
| Adjusted OR (95% CI) | Ref | 0.89 (0.74, 1.07) | 0.80 (0.63, 1.02) | 0.95 (0.47, 2.01) | 0.61 (0.30, 1.27) | 3.05 (0.56, 56.94) | 1.06 (0.78, 1.47) |
| Non-invasive ventilation, n (%) | 3104 (31.19) | 312 (35.74) | 163 (34.46) | 6 (14.63) | 10 (23.81) | 12 (70.59) | 121 (40.33) |
| Unadjusted OR (95% CI) | Ref | **1.25 (1.08, 1.45)** | 1.18 (0.97, 1.44) | **0.39 (0.15, 0.85)** | 0.70 (0.33, 1.38) | **5.40 (2.00, 17.00)** | **1.52 (1.20, 1.92)** |
| Adjusted OR (95% CI) | Ref | 1.09 (0.93, 1.28) | 0.98 (0.79, 1.21) | **0.41 (0.15, 0.97)** | 0.81 (0.37, 1.66) | **4.08 (1.46, 13.20)** | **1.35 (1.04, 1.74)** |
| Invasive ventilation, n (%) | 933 (9.38) | 125 (14.32) | 79 (16.70) | 1 (2.44) | 7 (16.67) | 4 (23.53) | 34 (11.33) |
| Unadjusted OR (95% CI) | Ref | **1.71 (1.39, 2.09)** | **2.05 (1.59, 2.63)** | 0.26 (0.01, 1.18) | 2.05 (0.83, 4.35) | 3.15 (0.89, 8.92) | 1.31 (0.89, 1.86) |
| Adjusted OR (95% CI) | Ref | **1.68 (1.36, 2.06)** | **1.97 (1.51, 2.54)** | 0.26 (0.01, 1.20) | 2.08 (0.84, 4.45) | 2.87 (0.80, 8.18) | 1.34 (0.91, 1.91) |

Adjusted odds ratios are from multivariate regression analyses with age at admission, diabetes, obesity, SARS-CoV-2 variant era, remdesivir use, and Paxlovid use as covariates with non-exposure (without immunosuppression) as the reference. Significant odds ratios are in bold.

patients with PI (adjusted OR: 4.08, 95% CI 1.46–13.20) and patients with SI (adjusted OR: 1.35, 95% CI: 1.04–1.74) had significantly higher non-invasive ventilation than patients without immunosuppression. In addition, PWH without low CD4 had significantly lower non-invasive ventilation (adjusted OR: 0.41, 95% CI: 0.15–0.97) compared to patients without immunosuppression (Table 4).

Invasive ventilation was significantly higher in patients with immunosuppression compared to patients without immunosuppression in both unadjusted (OR: 1.71, 95% CI: 1.39–2.09) and adjusted (adjusted OR: 1.68, 95% CI: 1.36–2.06) analyses. The same trend was observed in patients with SOT (adjusted OR: 1.97, 95% CI: 1.51–2.54). No other subgroups had significantly different invasive ventilation compared to patients without immunosuppression.

 

## Length of hospitalization and ICU stay

In addition to the binary outcomes discussed above, we analyzed differences in hospitalization length and ICU length of stay among the groups. In linear regression analyses, patients with immunosuppression had significantly longer hospitalization length (median: 7 days) compared to patients without immunosuppression (median: 5 days, adjusted p-value < 0.001) when adjusting for age at admission, diabetes, obesity, SARS-CoV-2 variant era, and COVID-19 medication use. This remained true when stratifying hospitalization length into length to discharge alive and length to in-hospital mortality. ICU length of stay was also significantly longer in this group (median: 2.5 days in patients with immunosuppression, 2 days in patients without immunosuppression, adjusted p-value = 0.04). However, when stratifying by discharge disposition, only ICU length of stay to in-hospital mortality was significantly longer. Subgroup analysis showed the same trends for patients with SOT (adjusted p-value hospitalization length < 0.001, adjusted p-value ICU length of stay = 0.005). On the other hand, PWH without low CD4 had significantly shorter length of ICU stay (median: 1 day) compared to patients without immunosuppression (median: 2 days, adjusted p-value = 0.02). However, when stratifying by discharge disposition, only ICU length of stay was significantly shorter. PWH with low CD4 had no significant differences in hospitalization length or ICU length of stay (Table 5, S4 Table).

To account for the competing risk of in-hospital mortality on hospitalization length and ICU length of stay, we also compared the cumulative incidence functions of discharge between exposure groups using Gray's test and performed Fine-Gray regression analyses of these outcomes. Gray's test showed that the cumulative incidence functions for hospital discharge and ICU discharge were significantly different between groups (p < 0.001 for hospital discharge, p = 0.013 for ICU discharge). When accounting for competing risks, patients with immunosuppression still had longer hospitalization

**Table 5. Hospitalization length and ICU length of stay of the COVID-19 study population across groups.**

| | Non-exposure | Exposure | Solid organ transplant | HIV without low CD4 | HIV with low CD4 | Primary immunodeficiency | Secondary immunodeficiency |
|---|---|---|---|---|---|---|---|
| | n = 9079 | n = 873 | n = 473 | n = 41 | n = 42 | n = 17 | n = 300 |
| **Median hospitalization length (IQR)** | 5.00 (3.00, 10.00) | 7.00 (4.00, 15.00) | 7.00 (4.00, 15.0) | 5.00 (3.00, 11.0) | 5.00 (3.00, 10.0) | 13.0 (5.00, 27.0) | 7.00 (4.00, 15.0) |
| **Unadjusted p-value** | Ref | **< 0.001** | **< 0.001** | 0.8614 | 0.597 | **< 0.001** | **< 0.001** |
| **Adjusted p-value** | Ref | **< 0.001** | **< 0.001** | 0.4108 | 0.3532 | **0.0012** | **< 0.001** |
| **Adjusted HR (95% CI)** | Ref | **0.74 (0.69, 0.79)** | **0.77 (0.70, 0.85)** | 0.94 (0.71, 1.24) | **0.65 (0.46, 0.91)** | 0.80 (0.59, 1.08) | **0.68 (0.60, 0.77)** |
| | n = 2757 | n = 335 | n = 191 | n = 9 | n = 16 | n = 4 | n = 115 |
| **Median ICU length of stay (IQR)** | 2.00 (1.00, 5.00) | 2.50 (1.00, 7.00) | 2.00 (1.00, 8.00) | 1.00 (0.93, 2.00) | 3.00 (0.99, 6.00) | 13.00 (6.75, 18.25) | 3.00 (1.00, 6.00) |
| **Unadjusted p-value** | Ref | **0.0069** | **< 0.001** | **0.0312** | 0.5402 | 0.2554 | 0.3485 |
| **Adjusted p-value** | Ref | **0.0439** | **0.0054** | **0.0228** | 0.3597 | 0.3353 | 0.4414 |
| **Adjusted HR (95% CI)** | Ref | **0.79 (0.69, 0.91)** | **0.78 (0.66, 0.94)** | 1.68 (0.58, 4.87) | 0.71 (0.37, 1.34) | **1.43 (1.16, 1.77)** | **0.75 (0.59, 0.95)** |

Adjusted p-values are from multivariate regression analyses with age at admission, diabetes, obesity, SARS-CoV-2 variant era, remdesivir use, and Paxlovid use as covariates with non-exposure (without immunosuppression) as the reference. p-values < 0.05 are significant (bold). Adjusted hazard ratios are from Fine-Gray regression analyses with age at admission, diabetes, obesity, SARS-CoV-2 variant era, remdesivir use, and Paxlovid use as covariates with in-hospital mortality as the competing risk. Significant hazard ratios are in bold.

length (adjusted HR: 0.74, 95% CI: 0.69–0.79). This was also true for ICU length of stay (adjusted HR: 0.79, 95% CI: 0.69–0.91). Subgroup analysis still showed the same trends for patients with SOT when accounting for competing risks (adjusted HR hospitalization length: 0.77, 95% CI: 0.70–0.85; adjusted HR ICU length of stay: 0.78, 95% CI: 0.66–0.94). However, PWH with low CD4 also had longer hospitalization length when accounting for competing risks (adjusted HR: 0.65, 95% CI: 0.46–0.91). In addition, PWH without low CD4 did not have a significantly different ICU length of stay when accounting for competing risks (Table 5).

## Discussion

The objectives of this study were to compare in-hospital mortality, ICU admission, oxygenation requirements, and hospital/ICU length of stay among adults without immunosuppression and adults with different reasons for immunosuppression (SOT, HIV, PI, and SI) hospitalized with COVID-19. We identified 9952 adult patients hospitalized with COVID-19 for at least two days at NM hospitals between 03/01/2020 and 05/31/2022, and we categorized 873 of these patients as having a pre-existing form of immunosuppression. The proportions of individuals found to have each form of immunosuppression of interest followed our expectations based on the estimated prevalence of these conditions in the US and the fact that we were likely to oversample for such conditions given NM is an academic health system [33,34]. The differences observed in mean age and sex distribution for PWH compared to the total study population and other groups were expected based on the known demographics and age distribution for PWH [35].

Overall, the exposure group (with immunosuppression) had significantly worse outcomes than the non-exposure group (without immunosuppression), with higher in-hospital mortality, ICU admission, invasive ventilation, and longer hospital/ICU length of stay. When dividing those with immunosuppression into subgroups, patients with SOT had the poorest experiences following hospitalization with COVID-19, performing significantly worse than patients without immunosuppression on all the same outcomes. This aligns with a systematic review of patients with SOT compared to patients from the general population hospitalized with COVID-19 which also showed higher mortality and rate of intensive care in patients with SOT [1]. However, while this systematic review found higher mortality in patients with SOT in unadjusted analyses, we observed no differences in mortality when we did not adjust for comorbidities such as age, diabetes, and obesity.

In-hospital mortality, ICU admission, and hospitalization length accounting for competing risks were significantly higher/longer in PWH with low CD4. A cohort study in Italy of PWH with low CD4 T cell count hospitalized with COVID-19 also found higher mortality and significantly longer hospitalization in this group compared to the general population. Notably, we did not observe any significant differences in oxygenation requirements or length of ICU stay compared to patients without immunosuppression [25]. It was interesting that we observed a similar trend in mortality and hospitalization length despite other outcomes that were otherwise not worse compared to patients without immunosuppression. This could be due to low CD4 T cell count being a marker for less engagement in care or due to these patients dying from something other than COVID-19. Unlike the other immunosuppression subgroups, PWH without low CD4 T cell count had similar or better outcomes than patients without immunosuppression. This was expected as these patients were likely well engaged in care including consistent use of combination antiretroviral therapy that led to great immune recovery.

Patients with SI had significantly higher in-hospital mortality, ICU admission, non-invasive ventilation, and longer hospitalization length and ICU length of stay accounting for competing risks compared to patients without immunosuppression. While patients with PI also had higher non-invasive ventilation, mortality and ICU admission were not significantly different from patients without immunosuppression. This aligns with findings from a systematic review performed by the CDC showing greater morbidity for patients with PI who develop COVID-19 [9] and previous results from a cohort study in the United Kingdom Primary Immunodeficiency Network (UKPIN) that showed no differences in mortality for patients with PI compared to patients without immunosuppression [7]. However, it is important to note that the UKPIN study examined patients who developed COVID-19 and not necessarily patients hospitalized with COVID-19. Overall, our results

add knowledge of the relationship between PI and SI and COVID-19 by specifically examining patients hospitalized with COVID-19, which no other studies have done in the context of PI or SI.

Patients with SOT may have particularly poor outcomes due to long-term higher levels of immunosuppression from medications to minimize organ rejection. Corticosteroids, which suppress general inflammation, and calcineurin inhibitors, which suppress T cell development, are most commonly employed. In contrast, PWH experience a loss of adaptive immunity, and PI and SI can lead to a loss of adaptive or innate immunity based on the specific immunodeficiency a patient has. Clearly, different mechanisms of immunosuppression can result in varying severity and outcomes of COVID-19 hospitalization. However, it is important to keep in mind that both our groups of PWH and patients with PI were small, and we may not be adequately powered to detect true differences.

Our study is also limited by potential data missingness and misclassification inherent to EHR data. Firstly, because the EHR is a real-world data source, it is not guaranteed to capture all the desired information about a patient. For example, information can be fragmented across multiple systems and may not always be entered accurately. Thus, the presence or absence of information in the EHR may not always reflect reality regarding factors like immunosuppression status. In addition, while we manually adjudicated all patients in the immunosuppression subgroups, we did not exhaustively adjudicate all variables for the entire study population. Thus, because we only separated out patients who were coded as immunosuppressed in the EHR, we recognize that there may be patients in the without immunosuppression group who should be in one of the immunosuppression subgroups. This is especially true for our SI group, which represents a broad category of immunosuppression. This could potentially bias our non-exposure group toward appearing more severe than they are. However, the magnitude of bias is expected to be small given our comprehensive method of filtering for immunosuppression, the rarity of immunosuppression, and the large size of our cohort.

Selection bias is also important to consider in retrospective cohort studies, but there is less control over this form of bias in EHR-based studies, and it generally has less impact. In particular, our study broadly identifies as many patients as possible from EHR data. For example, as COVID-19 testing is mandatory for all patients admitted to NM hospitals, selection bias is minimized by making positive COVID-19 testing a key inclusion criterion. Furthermore, by only including patients who were hospitalized for at least 2 days, we avoided including individuals who may not have required hospitalization and kept our study focused on in-hospital outcomes.

Hoewver, our focus on hospitalization EHR data and not assessing outcomes beyond discharge such as hospital readmission or death outside of the hospital setting, is also a limitation. In addition, given the irregularity of EHR data collection, another limitation is that it is difficult to consistently assess competing risks across all participants, so we did not use competing risks analysis when assessing the cumulative incidence of the binary outcomes. Thus, these outcomes may be vulnerable to effects on competing risks. For the hospitalization length and ICU length of stay outcomes, we accounted for competing risks by performing Fine-Gray regression analyses. Finally, our study is limited to a single health system and therefore may not be representative of the general population. This is especially important given NM is an academic health system with more complex, sicker, and better insured patients compared to community-based hospital systems.

Our next steps are to expand our study to include multi-health system EHR data. By applying the COVID-19 and immunosuppression definitions we developed at a larger population level, we can greatly increase our sample sizes for each group and gain a more representative sampling across the US. Because our present study only ascertained hospital outcomes, we also plan to study longer term outcomes such as one-year all-cause mortality or development of long COVID. This may provide further insight into the lasting effects of COVID-19, which can impact health well beyond hospital discharge [36].

## Conclusion

Adult patients with immunosuppression experience worse outcomes than adult patients without immunosuppression following hospitalization with COVID-19. Patients with SOT had the poorest outcomes, including higher in-hospital mortality,

ICU admission, invasive ventilation, and longer hospital/ICU stay. PWH with low CD4 and patients with SI also had higher in-hospital mortality and ICU admission. These findings underscore the importance of tailored approaches in managing COVID-19 among populations with immunosuppression. Future cohort studies should aim to expand on these results by incorporating multi-health system data to increase sample sizes and improve representativeness. Additionally, investigating long-term outcomes could provide crucial insights into the lasting effects of COVID-19 on individuals with immunosuppression. Such comprehensive research will be invaluable in guiding healthcare providers and public health professionals in their ongoing efforts to protect vulnerable populations during this and future pandemics.

## Supporting information

**S1 Fig. Flowchart of study population identification and grouping.** From 03/01/2020 to 05/31/2022, 10713 adult patients were hospitalized with COVID-19 at Northwestern Medicine. Of these patients, 9952 were hospitalized for at least two days and were categorized as without immunosuppression or with immunosuppression. Those with immunosuppression were further categorized as solid organ transplant, HIV, primary immunodeficiency, or secondary immunodeficiency based on the presence of diagnosis codes and manual adjudication. People with HIV were further categorized as without low CD4 or with low CD4 based on their absolute CD4 cell count closest in time to hospitalization.
(TIF)

**S1 Table. Full regression analysis results.**
(DOCX)

**S2 Table. Sex-stratified results, male.**
(DOCX)

**S3 Table. Sex-stratified results, female.**
(DOCX)

**S4 Table. Hospitalization length and ICU length of stay stratified by discharge disposition.**
(DOCX)

**S1 File. STROBE guidelines checklist.**
(DOCX)

## Author contributions

**Conceptualization:** Vijeeth Guggilla, Jennifer A Pacheco, Alexandre M Carvalho, Chad J Achenbach, Theresa L Walunas.

**Data curation:** Vijeeth Guggilla, Jennifer A Pacheco, Anna E Pawlowski.

**Formal analysis:** Vijeeth Guggilla.

**Funding acquisition:** Theresa L Walunas.

**Investigation:** Vijeeth Guggilla, Jennifer A Pacheco.

**Methodology:** Vijeeth Guggilla, Jennifer A Pacheco, Alexandre M Carvalho, Chad J Achenbach, Theresa L Walunas.

**Project administration:** Theresa L Walunas.

**Resources:** Theresa L Walunas.

**Software:** Vijeeth Guggilla, Jennifer A Pacheco.

**Supervision:** Chad J Achenbach, Theresa L Walunas.

**Validation:** Vijeeth Guggilla, Alexandre M Carvalho, Grant R Whitmer, Chad J Achenbach, Theresa L Walunas.

**Visualization:** Vijeeth Guggilla.

**Writing – original draft:** Vijeeth Guggilla.

**Writing – review & editing:** Vijeeth Guggilla, Jennifer A Pacheco, Alexandre M Carvalho, Grant R Whitmer, Anna E Pawlowski, Jodi L Johnson, Catherine A Gao, Chad J Achenbach, Theresa L Walunas.

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
