## [Decision Letter · Decision Letter 0]

9 Jan 2025

Dear Dr. Walunas,

Thank you for submitting your manuscript to PLOS ONE. After careful consideration, we feel that it has merit but does not fully meet PLOS ONE’s publication criteria as it currently stands. Therefore, we invite you to submit a revised version of the manuscript that addresses the points raised during the review process.

We look forward to receiving your revised manuscript.

Kind regards,

Mickael Essouma, M. D.

Academic Editor

PLOS ONE

Journal Requirements:

“TLW received grant number CO-US-540-6535 from Gilead Sciences (https://www.gilead.com/) for this work.

The funders reviewed the final manuscript.”

5. Please ensure that you refer to Figure 1 in your text as, if accepted, production will need this reference to link the reader to the figure.

6. Please upload a copy of Supporting Information Figure/Table/etc. SI (Fig 1) which you refer to in your text on page 11.

Reviewers' comments:

Reviewer's Responses to Questions

**Comments to the Author**

1. Is the manuscript technically sound, and do the data support the conclusions?

Reviewer #1: Yes

Reviewer #2: Yes

Reviewer #3: Yes

2. Has the statistical analysis been performed appropriately and rigorously?

Reviewer #1: Yes

Reviewer #2: Yes

Reviewer #3: No

3. Have the authors made all data underlying the findings in their manuscript fully available?

Reviewer #1: Yes

Reviewer #2: Yes

Reviewer #3: Yes

4. Is the manuscript presented in an intelligible fashion and written in standard English?

Reviewer #1: Yes

Reviewer #2: Yes

Reviewer #3: Yes

Reviewer #1: The study was well designed and organised providing important results that represent a valuable contribution to our COVID-19 knowledge. The article is well written and is suitable for publishing.

The only part that might be considered to be omitted from the manuscript is data about the patients' insurance as this does not share any important information in this research.

Reviewer #2: The authors were studying the differences in the outcomes of hospitalized patients with covid 19 with pre-existing immunosuppressive conditions.

This is an interesting study. The authors have clearly indicated the need for conducting the study. The methodology is also clear.

However, there are a few clarifications they need to make;

1. Results section: Hospitalization outcomes; it is not clear whether the hospitalization mentioned here includes ICU hospitalization. This has also to be clarified in table 2 since there is another table i.e Table 3 with ICU admissions.

2. Discussion: Check paragragh 4, While the authors mention that they did not observe any significant difference in length of stay in ICU for PWH with low CD4 counts, table 3 and the results section show otherwise.

Reviewer #3: Summary of the study.

The study seeks to investigate whether the adults with SOT, HIV , Primary and secondary immunodeficiency develop more severe COVID-19 than immunocompetent adults. The study seeks to discern more about the outcomes for adults with these conditions in adults hospitalized with COVID-19.

To address the research question, the authors conducted a retrospective study on COVID-19 hospitalized patients from 03/01/2020 to 05/31/2022 and using regression analysis, comparisons in hospitalization length, oxygenation requirements, ICU admission/length of stay and in-hospital mortality between patients with SOT,HIV with low CD4 cent count, HIV without low CD4 cell count, PI, SI against immunocompetent adults.

The literature cites strong findings demonstrating increased severe outcomes in these immunocompromised cohorts compared to the immunocompetent COVID-19 hospitalized patients; however, several studies have also reported no difference between immunocompromised and immunocompetent cohorts. The author’s main findings are patients with SOT, PI, SI had worse outcomes than immunodeficient patients following COVID-19 hospitalization.

Introduction: The study describes the evolution of COVID-19 pandemic and focusses in the severe COVID-19 outcomes, underscoring the elevated risk for the outcomes in pre-existing immunosuppressive conditions as this still remains less understood.

Specifically, no studies comparing outcomes of COVID-19 hospitalization have been reported, no students investigating the influence of low CD4 cell count in PWH on COVID-19 hospitalization outcomes have been reported. Other than mortality the other COVID-19 hospitalization outcomes have been reported in fewer studies.

The study identifies these gaps and seeks to add to knowledge of COVID-19 epidemiology using data sourced from EHR for a single-health system. Difference in hospitalization length, oxygenation requirements, ICU admission/length of stay, and in-hospital mortality between COVID-19 hospitalized patients with history of SOT, HIV, PI and SI compared to COVID-19 hospitalized patients with no history of immunosuppression.

Method

The authors performed a retrospective review of a single-health system data from the Northwestern Medicine Enterprise Data Warehouse. COVID-19 hospitalized patients included adults (at least 18 years of age) hospitalized with COVID-19 at NM hospitals between 03/01/2020 and 05/31/2022. Patients that ever received a positive COVID-19 laboratory test and were also hospitalized in the period encompassing one day before to seven days after the laboratory test result.

Data from EHR: Age at admission, sex assigned at birth, race, ethnicity, insurance status, and most recent body mass index (BMI) measurement. History of diabetes identified via ICD-9 or ICD-10. Dates of hospital and ICU admission and discharge, in-hospital oxygenation information, and in-hospital all-cause mortality were also extracted from the EHR. Dates of hospital admission were used to assign SARS-CoV-2 variant era with categorization into three groups(Pre-Delta, Delta, Omicron) based on calendar time of when a specific variant of concern (VOC) accounted for greater than 50% of circulating viruses determined per sequencing surveillance.

Statistical Consideration

Multivariate linear regression analyses adjusting for age at admission, diabetes, obesity, and SARS-CoV-2 variant era as covariates were performed to assess the effect of immune group status on mean 10 hospitalization length and mean length of ICU stay. For hospitalization length, the logarithm transform of the data was used to achieve normality of the residuals.

Please consider non-parametric methods: log transformation in the study is not suitable for two reasons. Firstly, the summary statistics on length of stay are presented as mean(SD) which is not appropriate when the data is skewed; consider summary stats for non-normal data; secondly, the log transformation of LOS in the regression analysis was motivated by need to have normality of the distribution of residuals. This would require back-transformation of estimates from “log” residual model . It is not clear how this was accomplished in the is study.

Multivariate Regression

Adjust for these covariates because age, diabetes, and obesity have all been identified as the strongest risk factors for death in patients hospitalized with COVID-19, and because of changes in pandemic landscape over time with the rise and fall of different VOCs.

TTE analysis: Fourteen and 28-day cumulative incidence of ICU admission and all-cause mortality censored at discharge were also assessed.

1) please state if there were the competing risks to compute cumulative incidence of ICU admission and all-cause mortality censored at discharge

2) Clarify the part “and all-cause mortality censored at discharge.”

3) For all logistic regression analyses, influential values were assessed and determined not to impact results. There was no multicollinearity observed across all the analyses.

4) Would the authors provide more information how influential values and multicollinearity were assessed?]

Table 1

The table presents descriptive statistic of age, sex, Race, ethnicity , Insurance status, SARS-CoV-2 variant era, BMI and Diabetes with frequency(percentage) for categorical data, as appropriate. Age of patients is presented as mean(SD). This is recommended when data is assumed to be normally distributed, however, there does not seem to be validation of this assumption. The authors should provide clarification support for the choice of mean(SD) for age.

Table 2.

Hospitalization Outcomes are presented in Table 2 includes summary of continuous variables hospitalization length, and comparison between no immunosuppression and the immunosuppression cohorts with age at admission, diabetes, obesity, and SARS-CoV-2 variant era as covariates. Hospitalization length was presented as mean(SD) , while the multivariate regression analysis also recognized that it is non-normality of the residuals by applying log transformation. This would require back-transformation of estimates from “log” residual model . Further, the adequacy of transformation was not discussed. It is not clear how this was accomplished in the is study. The table also presents frequency (percentage) of binary outcomes and p-values for testing their corresponding comparisons between immunosuppression cohort with no immunosuppression cohort using multivariate logistic regression analyses were performed adjusting for the same covariates.

Table 3.

ICU Outcomes are presented in Table 2 includes summary of continuous variables ICU admission length, and comparison between no immunosuppression and the immunosuppression cohorts with age at admission, diabetes, obesity, and SARS-CoV-2 variant era as covariates. ICU admission length was presented as mean(SD) , while the multivariate regression analysis also recognized that it is non-normality of the residuals by applying log transformation. This would require back-transformation of estimates from “log” residual model . Further, the adequacy of transformation was not discussed. It is not clear how this was accomplished in the is study. The table also presents frequency (percentage) of ICU admission and p-values for testing comparisons between immunosuppression cohort with no immunosuppression cohort using multivariate logistic regression analyses were performed adjusting for the same covariates.

Other Considerations

The comparison of outcomes between the immunosuppression cohorts was appropriate for addressing the research questions, notwithstanding the sample size for the HIV with low CD4 cell count, HIV without low CD4 cell count and primary immunodeficiency were limited 42, 41 and 17. The authors acknowledge the limitation of a single health system and that larger sample sizes may gain a more representative sampling across the US. While most of the methodology was described to facilitate reproducibility, additional details on how assumptions on normality of data, appropriateness/adequacy of log-transformed length of stay models using multivariate regression would enhance the ability of other research to reproduce the study. Cumulative incidence of binary outcomes at 14 day and 28 day timepoint may be vulnerable to effects on competing risks. This was not addressed, and neither was it discussed as a limitation.

Discussion:

The study provides explanation of differences in prevalence for the immunosuppression conditions and age of patients between US and the NM health system. The study reported poor prognosis in the SOT in terms of hospitalization length, rates of invasive ventilation, higher rates in ICU admission, and length of ICU stay. Further, in general, poorer prognosis was reported in PI and SI, adding to knowledge base. While this study investigated the COVID-19 hospitalization outcomes in the PWP with low CD4 cell count and PWP without low CD4 cell count, it did recognize that both groups were small sized. In conclusion, the study findings suggest that patients with immunosuppression from various causes have different outcomes after hospitalization with COVID-19, suggesting implications for future responses to COVID-19 and future pandemics.

**Do you want your identity to be public for this peer review?** For information about this choice, including consent withdrawal, please see our Privacy Policy

Reviewer #1: No

Reviewer #2: **Yes: ** Zillah Moraa Malachi

Reviewer #3: **Yes: ** Themba Nyirenda

---

## [Author Response · Author response to Decision Letter 1]

12 Mar 2025

Please see the attached response to reviewers document for an easier to read and better formatted version of our response to review comments.

Response to Reviewers and Academic Editor

Thank you to everyone for their thoughtful comments and edits that have helped further strengthen this manuscript.

The original comments from the Reviewers and Academic Editor are in italics, and my responses are below each comment in regular text. When appropriate, line numbers/sections to refer to and specific changes made are available in blue text under each response. Important deletions are marked as “DELETED.” Line numbers are referring to those in “Revised Manuscript with Track Changes.”

The comments from Reviewers 1-3 are addressed first, and the comments from the Academic Editor are addressed afterward. Additional Journal Requirements are addressed last.

Response to Reviewer 1

Comment 1: “The only part that might be considered to be omitted from the manuscript is data about the patients' insurance as this does not share any important information in this research.”

We thank the reviewer for bringing up the fact that the insurance information does not add to this research. We have omitted the data about patients’ insurance status from Table 1 as suggested.

See Table 1 (line 384).

Response to Reviewer 2

Comment 1: “Results section: Hospitalization outcomes; it is not clear whether the hospitalization mentioned here includes ICU hospitalization. This has also to be clarified in table 2 since there is another table i.e. Table 3 with ICU admissions.”

We appreciate the request for clarification on these outcomes. All patients included in this study were hospitalized, but not all patients were also admitted to the ICU. Hospitalization length is defined as the total amount of time a patient spent in the hospital, which would also include the time they spent in the ICU if they were admitted to the ICU. The language in the Results and Methods section has been updated to make this clearer and strengthen the manuscript. The tables have also been updated to make it clear that ICU admission is a distinct outcome that occurs at different rates in the different groups.

Line 266: “For patients who were also admitted to the ICU, hospitalization length includes time spent in the ICU”

Line 426: “Our study only included hospitalized patients, and many of these patients were also admitted to the ICU while hospitalized”

See Table 3 (line 481).

Comment 2: “Discussion: Check paragraph 4, While the authors mention that they did not observe any significant difference in length of stay in ICU for PWH with low CD4 counts, table 3 and the results section show otherwise.”

We thank the reviewer for bringing up a potential discrepancy in our discussion. We double checked our reporting and discussion of ICU length of stay in PWH with low CD4 counts. The adjusted p-value for ICU length of stay for this group was 0.36 and therefore not significant, and this is reflected in Table 5. This is reported in our Results section where we state “PWH with low CD4 had no significant differences in hospitalization length or ICU length of stay.” This is also consistent with our Discussion section where we state “However, while this study found that PWH with low CD4 cell count were hospitalized for significantly longer, we did not observe any significant differences in hospitalization length, oxygenation requirements, or length of ICU stay compared to patients without immunosuppression.” Therefore, the Discussion is consistent with the Results and Table 3.

See Table 5 (line 552)

Line 543: “PWH with low CD4 had no significant differences in hospitalization length or ICU length of stay”

Line 592: “However, while this study found that PWH with low CD4 cell count were hospitalized for significantly longer, we did not observe any significant differences in hospitalization length, oxygenation requirements, or length of ICU stay compared to patients without immunosuppression”

Response to Reviewer 3

Comment 1: “Please consider non-parametric methods: log transformation in the study is not suitable for two reasons. Firstly, the summary statistics on length of stay are presented as mean(SD) which is not appropriate when the data is skewed; consider summary stats for non-normal data; secondly, the log transformation of LOS in the regression analysis was motivated by need to have normality of the distribution of residuals. This would require back-transformation of estimates from “log” residual model . It is not clear how this was accomplished in the is study.”

We appreciate the reviewer for considering the best methods to present summary stats and for asking us to elaborate on our analyses. We agree that mean(SD) may not be appropriate when the data is skewed, and we have accordingly changed all the summary stats to reflect median and interquartile range (IQR) instead as these are the appropriate summary statistics to use when the normality of data is not explicitly validated. We have also strengthened the manuscript by adding explanation in the methods section of how back-transformation of estimates from the log residual model were accomplished using exponentiation via the emmeans R package.

See Table 1 (line 384).

See Table 5 (line 552).

Line 315: “For hospitalization length, the logarithm transform of the data was used to achieve normality of the residuals and back-transformation of estimates from the log residual model were produced using exponentiation via the emmeans R package”

Comment 2: “please state if there were the competing risks to compute cumulative incidence of ICU admission and all-cause mortality censored at discharge”

We thank the reviewer for bringing this into consideration. Because we are using retrospective EHR data, which is irregularly sampled, we cannot consistently assess competing risks across all participants. As the reviewer suggests, this is an important limitation to mention. We have added a statement in the Methods section that competing risks were not used when analyzing the incidence of ICU admission and all-cause mortality. We have also mentioned this as a limitation in the Discussion section and added further context on why it is a limitation of our data source.

Line 305: “We did not apply competing risks analysis when assessing these binary variables”

Line 673: “In addition, given the irregularity of EHR data collection, another limitation is that it is difficult to consistently assess competing risks across all participants, so we did not use competing risks analysis when assessing the cumulative incidence of the binary outcomes. Thus, these outcomes may be vulnerable to effects on competing risks”

Comment 3: “Clarify the part “and all-cause mortality censored at discharge.”

We appreciate the reviewer’s request for clarification. We had meant that patients were classified as surviving at time points that occurred after they were discharged alive. Now that we have simplified our outcomes based on other comments/feedback to look at overall in-hospital mortality and not mortality at different time points, this statement is no longer necessary, so we have removed it. The assessment of mortality should be much clearer overall now.

See Table 2 (line 419).

Line 290: DELETED “Fourteen and 28-day cumulative incidence of ICU admission and all-cause mortality censored at discharge were also assessed”

Comment 4: “Would the authors provide more information how influential values and multicollinearity were assessed?”

We thank the reviewer for finding ways to strengthen the Methods of our manuscript. Influential values were assessed by isolating any observations with a standardized residual greater than three and confirming that removal of these observations did not impact the results. Multicollinearity was assessed using the variance inflation factor. This information has been added to the Methods section, further strengthening the manuscript.

Line 306: “For all logistic regression analyses, influential values were assessed by isolating any observations with a standardized residual greater than three and performing sensitivity analyses. The variance inflation factor was used to confirm that there was no multicollinearity observed across all the analyses”

Comment 5: “Age of patients is presented as mean(SD). This is recommended when data is assumed to be normally distributed, however, there does not seem to be validation of this assumption. The authors should provide clarification support for the choice of mean(SD) for age.”

We appreciate the reviewer’s thorough perusal of our summary stats. We have changed all the summary stats (including that for age) to reflect median and interquartile range (IQR) instead of mean(SD) as these are the appropriate summary statistics to use when the normality of data is not explicitly validated.

See Table 1 (line 384).

Comment 6: “Hospitalization length was presented as mean(SD) , while the multivariate regression analysis also recognized that it is non-normality of the residuals by applying log transformation. This would require back-transformation of estimates from “log” residual model . Further, the adequacy of transformation was not discussed. It is not clear how this was accomplished in the is study.”

We appreciate the reviewer’s thorough perusal of our summary stats and gaps in our log residual model explanation. We have changed all the summary stats (including that for hospitalization length) to reflect median and interquartile range (IQR) instead of mean(SD) as these are the appropriate summary statistics to use when the normality of data is not explicitly validated. We also added explanation in the methods section of how back-transformation of estimates from the log residual model were accomplished using exponentiation via the emmeans R package.

See Table 5 (line 552).

Line 315: “For hospitalization length, the logarithm transform of the data was used to achieve normality of the residuals and back-transformation of estimates from the log residual model were produced using exponentiation via the emmeans R package”

Comment 7: “ICU admission length was presented as mean(SD) , while the multivariate regression analysis also recognized that it is non-normality of the residuals by applying log transformation. This would require back-transformation of estimates from “log” residual model . Further, the adequacy of transformation was not discussed. It is not clear how this was accomplished in the is study.”

We appreciate the reviewer’s thorough perusal of our summary stats. While log transformation was applied for hospitalization length, it was not needed for ICU admission length. Nonetheless, we have changed all the summary stats (including that for ICU admission length) to reflect median and interquartile range (IQR) instead of mean(SD) as these are the appropriate summary statistics to use when the normality of data is not explicitly validated.

See Table 5 (line 552).

Comment 8: “While most of the methodology was described to facilitate reproducibility, additional details on how assumptions on normality of data, appropriateness/adequacy of log-transformed length of stay models using multivariate regression would enhance the ability of other research to reproduce the study. Cumulative incidence of binary outcomes at 14 day and 28 day timepoint may be vulnerable to effects on competing risks. This was not addressed, and neither was it discussed as a limitation.”

We sincerely appreciate the reviewer’s detailed suggestions to improve discussion of our methodology. As suggested, we have removed any assumptions on normality of data (opting for non-parametric summary statistics of median and IQR), and we have further explicated the use of a log-transformed model for hospital length of stay and the appropriate back-transformations that were performed. Regarding the feedback that binary outcomes at 14 day and 28 day timepoints may be vulnerable to effects on competing risks, we thought about this carefully and decided that it would be more useful to simply focus on the outcomes during all of hospitalization instead of trying to assess outcomes at these time points. This would also be less vulnerable to competing risks. Thus, we have changed our binary outcomes to reflect occurrence during all of hospitalization. We have also addressed the fact that we did not use competing risks analysis for our binary outcomes, and we now discuss this as a limitation in the Discussion section.

See Table 1 (line 384).

See Table 5 (line 552).

Line 315: “For hospitalization length, the logarithm transform of the data was used to achieve normality of the residuals and back-transformation of estimates from the log residual model were produced using exponentiation via the emmeans R package”

Line 259: “The primary outcome was in-hospital mortality. Secondary outcomes included hospitalization length, oxygenation requirements, and ICU admission/length of stay”

See Table 2 (line 419).

See Table 3 (line 481).

See Table 4 (line 527).

Line 305: “We did not apply competing risks analysis when assessing these binary variables”

Line 673: “In addition, given the irregularity of EHR data collection, another limitation is that it is difficult to consistently assess competing risks across all participants, so we did not use competing risks analysis when assessing the cumulative incidence of the binary outcomes. Thus, these outcomes may be vulnerable to effects on competing risks”

Response to Academic Editor

Comment 1: I have revised this title like this because their immunocompromising status is a life-long condition, not just a "pre-COVID" condition. Furthermore, solid organ transplant is not an immunosuppressive condition, but patients who underwent solid organ transplantation are considered individuals with an immunocompromising status mainly because of their concomitant treatment with immunosuppressants. Because "hospitalization" for COVID-19 is a study outcome, the word "hospitalized" needs to be removed from the manuscript's title.

We thank the editor for their suggestions on improving the title. To provide clarification, the intent of this study was to assess post-hospitalization outcomes in patients who were hospitalized with COVID-19, and all 9952 patients in this study were hospitalized with COVID-19. Thus, hospitalization is not a study outcome, but rather one of the inclusion criteria for this study. We have clarified this in the Methods section description of inclusion criteria, and we are therefore inclined to keep the word “hospitalized” in the manuscript title as it describes an important characteristic of the population in this study. As for the distinction that immunosuppression is not just a condition, we agree with your assessment and have updated the title accordingly.

Line 1: “Immunosuppression variably impacts outcomes for patients hospitalized with COVID-19: A retrospective cohort study”

Comment 2: If the NM health system database is representative of a US population, you could say "retrospective population-based cohort study"

We appreciate the editor considering this as a possibility. However, as we mention in the Discussion section, one limitation of our study is that NM is an academic health system with more complex, sicker, and better insured patients compared to community-based hospital systems, meaning it may not be fully representative of the general population. Therefore, we think it is best to keep this as “retrospective cohort study.”

Line 677: “Finally, our study is limited to a single health system and therefore may not be representative of the general population. This is especially important given NM is an academic health system with more complex, sicker, and better insured patients compared to community-based hospital systems”

Line 1: “Immunosuppression variably impacts outcomes for patients hospitalized with COVID-19: A retrospective cohort study”

Comment 3: Consider numbering lines of the manuscript to ease its assessment.

We have numbered the lines of the manuscript as suggested.

Comment 4: Maximum word count = 300

We have kept the Abstract to less than 300 words as instructed.

Comment 5: I am happy that you use the word "adult" here, but it should also be frequently used thr

---

## [Decision Letter · Decision Letter 1]

9 Apr 2025

Dear Dr. Walunas,

Thank you for submitting your manuscript to PLOS ONE. After careful consideration, we feel that it has merit but does not fully meet PLOS ONE’s publication criteria as it currently stands. Therefore, we invite you to submit a revised version of the manuscript that addresses the points raised during the review process.

We look forward to receiving your revised manuscript.

Kind regards,

Mohammad Barary

Academic Editor

PLOS ONE

Reviewers' comments:

Reviewer's Responses to Questions

**Comments to the Author**

Reviewer #1: All comments have been addressed

Reviewer #2: All comments have been addressed

Reviewer #3: (No Response)

2. Is the manuscript technically sound, and do the data support the conclusions?

Reviewer #1: Yes

Reviewer #2: Yes

Reviewer #3: Yes

3. Has the statistical analysis been performed appropriately and rigorously?

Reviewer #1: Yes

Reviewer #2: Yes

Reviewer #3: I Don't Know

4. Have the authors made all data underlying the findings in their manuscript fully available?

Reviewer #1: Yes

Reviewer #2: Yes

Reviewer #3: Yes

5. Is the manuscript presented in an intelligible fashion and written in standard English?

Reviewer #1: Yes

Reviewer #2: Yes

Reviewer #3: Yes

Reviewer #1: The article is well structured and well written. It provides a certain contribution to our COVID-19 knowledge.

Reviewer #2: ICU hospitalisation has been clarified. The differences in table 2 and 3 are now clear as explained. The discrepancy in paragraph 4 in the discussion section has been addressed.

Reviewer #3: Thank you for taking the time to address all the issues raised in our last round of reviews. Primarily, the capture of discharge disposition allows the researcher to have a first step for determining competing risks. For example if we are interested in length of stay, then the time to event(TTE) of interest is time to being discharged alive, which makes in-hospital mortality(discharged not alive) a competing risk. Both of these variables should be available in the discharge disposition, in general. Further, one is not burdened with validating whether the Time to event follows a normal distribution or not. More importantly most of the endpoints of interest in this study are a variation of length of stay, which may by definition, i.e. duration from admission to discharge, focusses on the interval and ignores status of patient at discharge. So time to in-hospital mortality recognizes that intention to capture duration to the occurrence of death. As long as a patient admitted to the hospital or the ICU, and the possibility of death during the stay exists then for those that are not discharged alive, then the ability for evaluating the true LOS may have been terminated, I appreciate that this will be cited as a limitation, but it has not been explained which status of the patient at discharge can not be utilized to sort between the two types of length of stay, so that the researcher/reader is aware of how much bias of LOS will be present in the results of the current study, and showing the extent to which the findings are still valuable..

**Do you want your identity to be public for this peer review?** For information about this choice, including consent withdrawal, please see our Privacy Policy

Reviewer #1: No

Reviewer #2: **Yes: ** Zillah Moraa Malachi

Reviewer #3: **Yes: ** Themba Nyirenda

---

## [Author Response · Author response to Decision Letter 2]

9 May 2025

Response to Reviewers

Thank you to everyone for their comments on this second round of revisions.

The original comments from the Reviewers are in italics, and my responses are below each comment in regular text. When appropriate, line numbers/sections to refer to and specific changes made are available in blue text under each response. Important deletions are marked as “DELETED.” Line numbers refer to those in “Revised Manuscript with Track Changes.”

Response to Reviewer 1

Comment 1: “The article is well structured and well written. It provides a certain contribution to our COVID-19 knowledge.”

We thank the reviewer for taking the time to review this article.

Response to Reviewer 2

Comment 1: “ICU hospitalization has been clarified. The differences in table 2 and 3 are now clear as explained. The discrepancy in paragraph 4 in the discussion section has been addressed.”

We are glad these aspects are clear now. We thank the reviewer for taking the time to review this article.

Response to Reviewer 3

Comment 1: “Thank you for taking the time to address all the issues raised in our last round of reviews. Primarily, the capture of discharge disposition allows the researcher to have a first step for determining competing risks. For example, if we are interested in length of stay, then the time to event (TTE) of interest is time to being discharged alive, which makes in-hospital mortality(discharged not alive) a competing risk. Both of these variables should be available in the discharge disposition, in general. Further, one is not burdened with validating whether the Time to event follows a normal distribution or not. More importantly, most of the endpoints of interest in this study are a variation of length of stay, which may by definition, i.e. duration from admission to discharge, focuses on the interval and ignores status of patient at discharge. So time to in-hospital mortality recognizes that intention to capture duration to the occurrence of death. As long as a patient admitted to the hospital or the ICU, and the possibility of death during the stay exists then for those that are not discharged alive, then the ability for evaluating the true LOS may have been terminated, I appreciate that this will be cited as a limitation, but it has not been explained which status of the patient at discharge can not be utilized to sort between the two types of length of stay, so that the researcher/reader is aware of how much bias of LOS will be present in the results of the current study, and showing the extent to which the findings are still valuable.”

We appreciate the reviewer for emphasizing the importance of stratifying the length outcomes by discharge disposition to provide more granular information about the true lengths of stay for these patients (i.e., differentiating between time to in-hospital mortality and time to being discharged alive). Based on this feedback, we have performed an additional analysis of hospitalization length and ICU length of stay sorted by those who were eventually discharged and those who died before discharge, as suggested. This new analysis is discussed in the text (in the methods, results, and discussion section), and the full results are available in S4 Table. This will help the researcher/reader better interpret length of stay from this study and strengthen this article, and we thank the reviewer for their feedback.

Line 230: “For both hospitalization length and ICU length, stratified analyses of time to either discharge alive or time to in-hospital mortality were also performed to account for competing risks.”

Line 381: “This remained true when stratifying hospitalization length into length to discharge alive and length to in-hospital mortality.”

Line 385: “However, when stratifying by discharge disposition, only ICU length of stay to in-hospital mortality was significantly longer.”

Line 390: “However, there was no significant difference when we stratified by discharge disposition.”

Line 498: “For the hospitalization length and ICU length of stay outcomes, we accounted for competing risks by performing additional analyses stratifying by discharge disposition.”

See S4 Table (line 709).

---

## [Decision Letter · Decision Letter 2]

15 Jun 2025

Dear Dr. Walunas,

Thank you for submitting your manuscript to PLOS ONE. After careful consideration, we feel that it has merit but does not fully meet PLOS ONE’s publication criteria as it currently stands. Therefore, we invite you to submit a revised version of the manuscript that addresses the points raised during the review process.

We look forward to receiving your revised manuscript.

Kind regards,

Mohammad Barary, MD

Academic Editor

PLOS ONE

Journal Requirements:

Reviewers' comments:

Reviewer's Responses to Questions

**Comments to the Author**

Reviewer #3: (No Response)

2. Is the manuscript technically sound, and do the data support the conclusions?

Reviewer #3: No

3. Has the statistical analysis been performed appropriately and rigorously?

Reviewer #3: No

4. Have the authors made all data underlying the findings in their manuscript fully available?

Reviewer #3: Yes

5. Is the manuscript presented in an intelligible fashion and written in standard English?

Reviewer #3: Yes

Reviewer #3: Thank you for your time responding to comments previously raised on competing risks.

There is a methodology for handling competing risks. The recent review includes the strategy of presenting the analysis into two strata: those that die during the hospital stay and those that were discharged alive. The methodology evaluates cumulative incidence using cumulative incidence function (CIF) as opposed to usual KM curves which can not handle competing risks; and comparison of the CIF is conducted using Gray's test, instead the classing log-rank test. Since this methods accounts competing risks, the results of primary outcome of interests are presented on the entire data and not by stratum, which does not resolve the bias introduced by ignoring the competing risk structure of the occurrence of the events in this problem. Please consider employing the the unified approach for handling TTE in the presence of competing risk; i.e. death during hospitalization;

**Do you want your identity to be public for this peer review?** For information about this choice, including consent withdrawal, please see our Privacy Policy

Reviewer #3: **Yes: ** Themba Nyirenda

---

## [Author Response · Author response to Decision Letter 3]

1 Jul 2025

Response to Reviewer 3

Thank you for your time responding to comments previously raised on competing risks.

There is a methodology for handling competing risks. The recent review includes the strategy of presenting the analysis into two strata: those that die during the hospital stay and those that were discharged alive. The methodology evaluates cumulative incidence using cumulative incidence function (CIF) as opposed to usual KM curves which can not handle competing risks; and comparison of the CIF is conducted using Gray's test, instead the classing log-rank test. Since this methods accounts competing risks, the results of primary outcome of interests are presented on the entire data and not by stratum, which does not resolve the bias introduced by ignoring the competing risk structure of the occurrence of the events in this problem. Please consider employing the unified approach for handling TTE in the presence of competing risk; i.e. death during hospitalization.

We thank the reviewer for the additional feedback. As requested, we employed a unified approach for handling TTE using competing risks methodology. Discharge alive was treated as the primary event, with death during hospitalization as a competing risk. Analyses included cumulative incidence functions (CIFs) for discharge, Gray’s tests to compare discharge CIFs across exposure subgroups, and Fine-Gray proportional sub-distribution hazards models to quantify subgroup associations. This approach aligns with modern standards for time-to-event data with competing events.

Methods: “To account for competing risks when comparing hospitalization length and ICU length of stay, time-to-event analyses were also performed with discharge alive as the event of interest and in-hospital death as the competing event. Gray’s test was used to compare the cumulative incidence functions of discharge between exposure subgroups and a Fine-Gray proportional sub-distribution hazards model was implemented to examine the association between exposure subgroup and discharge incidence adjusting for the same covariates.”

Results: “To account for the competing risk of in-hospital mortality on hospitalization length and ICU length of stay, we also compared the cumulative incidence functions of discharge between exposure groups using Gray’s test and performed Fine-Gray regression analyses of these outcomes. Gray’s test showed that the cumulative incidence functions for hospital discharge and ICU discharge were significantly different between groups (p < 0.001 for hospital discharge, p = 0.013 for ICU discharge). When accounting for competing risks, patients with immunosuppression still had longer time to hospital discharge (adjusted HR: 0.74, 95% CI: 0.69-0.79). This was also true for ICU length of stay (adjusted HR: 0.79, 95% CI: 0.69-0.91). Subgroup analysis still showed the same trends for patients with SOT when accounting for competing risks (adjusted HR hospitalization length: 0.77, 95% CI: 0.70-0.85; adjusted HR ICU length of stay: 0.78, 95% CI: 0.66-0.94). However, PWH with low CD4 also had longer time to hospital discharge when accounting for competing risks (adjusted HR: 0.65, 95% CI: 0.46-0.91). In addition, PWH without low CD4 did not have a significantly different time to ICU discharge when accounting for competing risks (Table 5).”

See Table 5.

Table 5 Legend: “Adjusted hazard ratios are from Fine-Gray regression analyses with age at admission, diabetes, obesity, SARS-CoV-2 variant era, remdesivir use, and Paxlovid use as covariates with in-hospital mortality as the competing risk. Significant hazard ratios are in bold.”

Discussion: “In-hospital mortality, ICU admission, and hospitalization length accounting for competing risks were significantly higher/longer in PWH with low CD4. A cohort study in Italy of PWH with low CD4 cell count hospitalized with COVID-19 also found higher mortality and significantly longer hospitalization in this group compared to the general population. Notably, we did not observe any significant differences in oxygenation requirements or length of ICU stay compared to patients without immunosuppression [25]. It was interesting that we observed a similar trend in mortality and hospitalization length despite other outcomes that were otherwise not worse compared to patients without immunosuppression.”

---

## [Decision Letter · Decision Letter 3]

28 Jul 2025

Immunosuppression variably impacts outcomes for patients hospitalized with COVID-19: A retrospective cohort study

PONE-D-24-51256R3

Dear Dr. Walunas,

We’re pleased to inform you that your manuscript has been judged scientifically suitable for publication and will be formally accepted for publication once it meets all outstanding technical requirements.

Kind regards,

Mohammad Barary, MD

Academic Editor

PLOS ONE

Additional Editor Comments (optional):

Reviewers' comments:

Reviewer's Responses to Questions

**Comments to the Author**

Reviewer #3: All comments have been addressed

2. Is the manuscript technically sound, and do the data support the conclusions?

Reviewer #3: Yes

3. Has the statistical analysis been performed appropriately and rigorously?

Reviewer #3: Yes

4. Have the authors made all data underlying the findings in their manuscript fully available?

Reviewer #3: Yes

5. Is the manuscript presented in an intelligible fashion and written in standard English?

Reviewer #3: Yes

Reviewer #3: Thank you for taking the time to address the concerns with time-to-event in the presence of competing risks which I had previously raised. These have been adequately addressed in every relevant analysis :

" As requested, we employed a unified

approach for handling TTE using competing risks methodology. Discharge alive was

treated as the primary event, with death during hospitalization as a competing risk.

Analyses included cumulative incidence functions (CIFs) for discharge, Gray’s tests to

compare discharge CIFs across exposure subgroups, and Fine-Gray proportional subdistribution hazards models to quantify subgroup associations. This approach aligns

with modern standards for time-to-event data with competing events.

Methods: “To account for competing risks when comparing hospitalization length and

ICU length of stay, time-to-event analyses were also performed with discharge alive as

the event of interest and in-hospital death as the competing event. Gray’s test was used

to compare the cumulative incidence functions of discharge between exposure

subgroups and a Fine-Gray proportional sub-distribution hazards model was

implemented to examine the association between exposure subgroup and discharge

incidence adjusting for the same covariates.”

These changes have significantly enhanced the rigor of the methodology and interpretation or discussion of the findings.

Thank you again.

**Do you want your identity to be public for this peer review?** For information about this choice, including consent withdrawal, please see our Privacy Policy

Reviewer #3: **Yes: ** Themba Nyirenda

---

## [Editor Report · Acceptance letter]

PONE-D-24-51256R3

PLOS ONE

Dear Dr. Walunas,

I'm pleased to inform you that your manuscript has been deemed suitable for publication in PLOS ONE. Congratulations! Your manuscript is now being handed over to our production team.

Kind regards,

on behalf of

Dr. Mohammad Barary

Academic Editor

PLOS ONE